# The Impact of a Very-Low-Calorie Ketogenic Diet on Monocyte Subsets of Patients with Obesity: A Pilot Study

**DOI:** 10.3390/nu17020312

**Published:** 2025-01-16

**Authors:** Mariaignazia Curreli, Serena Recalchi, Davide Masi, Rebecca Rossetti, Ilaria Ernesti, Elisabetta Camaiani, Sabrina Basciani, Elena Gangitano, Mikiko Watanabe, Stefania Mariani, Lucio Gnessi, Stefania Morrone, Andrea Lenzi, Elisa Petrangeli, Carla Lubrano

**Affiliations:** Department of Experimental Medicine, Section of Medical Pathophysiology, Food Science and Endocrinology, Sapienza University of Rome, 00161 Rome, Italy

**Keywords:** obesity, monocytes, ketogenic diet, inflammation, VLCKD, flow cytometry

## Abstract

**Background/Objectives:** Obesity is closely linked to chronic low-grade inflammation and the development of cardio-metabolic comorbidities. Monocyte subsets, which are crucial in immune responses, have been reported to be altered in individuals with obesity, potentially exacerbating inflammation. Although very-low-calorie ketogenic diets (VLCKDs) are recognized for their efficacy in promoting weight loss and improving metabolic health, their impact on circulating monocyte subsets remains poorly understood. The objective of our study is to investigate the impact of VLCKDs on monocyte subset distribution in people with obesity. **Methods:** Thirty-six participants were divided into four groups—healthy controls, individuals with obesity and no dietary intervention, and individuals with obesity following either a low-calorie diet (LCD) or VLCKD for 28 days. Blood samples were analyzed to assess the distribution of classical monocytes (CMs), intermediate monocytes (IMs), and non-classical monocytes (NCMs) using flow cytometry. **Results:** Individuals with obesity exhibited significant increases in IMs and NCMs, alongside a decrease in CMs compared to healthy controls. The VLCKD led to a notable shift in monocyte distribution, with increased CMs and reduced IMs and NCMs, restoring levels closer to those observed in healthy individuals. In contrast, the LCD group showed no significant changes in monocyte subsets. **Conclusions:** VLCKDs may exert anti-inflammatory effects by modulating monocyte subset distribution, offering potential therapeutic benefits in mitigating obesity-related inflammation. These preliminary findings suggest that VLCKDs could be an effective strategy for improving immune function in individuals with obesity.

## 1. Introduction

Obesity is a significant driver of a broad spectrum of cardiometabolic complications, including type 2 diabetes mellitus (T2DM), hypertension (HTN), and metabolic-associated steatotic liver disease (MASLD). It also contributes to an increased risk and progression of several types of cancer, exacerbates autoimmune disorders, and increases susceptibility to infections. Furthermore, obesity has been shown to impair immune function, leading to a reduced responsiveness to vaccines [1,2]. A hallmark of central obesity and its associated disorders is low-grade chronic inflammation [3]. Individuals with obesity, especially those exhibiting a substantial accumulation of visceral fat, demonstrate heightened systemic levels of inflammatory markers, including C-reactive protein (CRP), various cytokines, and interleukins [4]. Furthermore, a robust strand of research has demonstrated that visceral fat expansion is characterized by adipocyte hypertrophy, limited cellular proliferation, and maladaptive dysfunction, which, together, promote a local inflammatory state that progresses into chronic systemic inflammation [5]. Adipocyte enlargement leads to hypoxia and initiates a stress response, driving the release of pro-inflammatory cytokines, pro-atherogenic adipokines, and chemoattractant proteins, ultimately resulting in monocyte recruitment and immune cell infiltration [6]. These microenvironmental changes significantly alter immune cell responses, as reflected by the altered proportions of circulating immune cells, including granulocytes, monocytes, and lymphocytes, which are closely associated with obesity and its complications [7]. Among these, circulating blood monocytes are particularly noteworthy, as they are considered a key source of macrophage accumulation in chronic obesity [8,9]. The role of monocytes in inflammation has been extensively documented in several pathologies [10]. In 2010, a group of experts proposed an official nomenclature for human blood monocyte subsets [11], categorized into three major subsets based on the expression of surface proteins CD14 (a glycosylphosphatidylinositol-anchored receptor that serves as a co-receptor for Toll-like receptors) and CD16 (the Fc gamma III receptor); the three subsets are classical monocytes (CMs; CD14++CD16−), non-classical monocytes (NCMs; CD14dimCD16+), and intermediate monocytes (IMs; CD14++CD16++) [11]. Although many studies have highlighted the functional differences between these monocyte subsets, a clear delineation of their distinct functional roles remains incomplete [12]. It is known that monocytes are released from the bone marrow as a homogeneous population of CD14++CD16− cells, some of which later differentiate into intermediate and non-classical monocytes [13]. Classical monocytes play a central role in phagocytosis and the generation of reactive oxygen species (ROS) as part of the immune response to bacterial infections and tissue injury. They are predominantly implicated in extravasation and migration to sites of injury and inflammation. In contrast, non-classical monocytes are involved in patrolling the vascular endothelium and clearing cellular debris, while intermediate monocytes excel in antigen presentation, cytokine secretion, apoptosis regulation, and cell differentiation. These cells have distinct chemokine expression profiles that may reflect different recruitment properties. In the context of obesity and its associated comorbidities, clinical studies have shown correlations between the number of blood monocytes and cardiovascular risk factors such as BMI, waist circumference, HDL cholesterol, and triglyceride levels [14]. Despite these insights, the precise roles of each monocyte subset in disease states remain incompletely understood [15]. Nonetheless, several studies have reported an increased proportion of circulating non-classical and intermediate monocytes in chronic inflammatory conditions, including obesity [13,16,17]. Emerging evidence suggests that a very-low-calorie ketogenic diet (VLCKD) is an effective intervention for obesity, inducing significant weight loss, exerting anti-inflammatory effects, and improving body composition and cardiometabolic outcomes [18,19,20,21]. However, data on the effects of VLCKDs on the inflammatory potential of circulating monocyte subsets are currently lacking. The aim of our study was to investigate the distribution and correlation of monocyte subsets with metabolic and inflammatory serum markers in individuals with obesity and typical-weight controls. Additionally, we sought to examine the impact of different dietary regimens, particularly VLCKDs, on monocyte subset distribution.

## 2. Materials and Methods

### 2.1. Study Design and Population

This single-center retrospective study was conducted at the high-specialization Center for the Treatment of Obesity (CASCO) at Sapienza University of Rome. The inclusion criteria for the study were as follows: age between 30 and 65 years, male or female, and a body mass index (BMI) >19 or <50 kg/m^2^. Exclusion criteria included the presence of acute or chronic inflammatory conditions, infectious diseases, cancer, significant alcohol consumption (>20 g/day), and contraindications for the VLCKD, such as type 1 diabetes, renal failure (eGFR < 60 mL/min/1.73 m^2^), liver failure (including decompensated cirrhosis), congenital metabolic disorders, pregnancy or lactation, and major psychiatric disorders.

Participants were divided into four groups. The first group comprised 9 healthy, typical-weight individuals who followed a balanced diet for 4 weeks prior to blood sample collection. The second group included 9 individuals with obesity who were not following any dietary restrictions. The third and fourth groups consisted of 18 individuals with obesity undergoing dietary interventions—9 followed a low-calorie diet (LCD) for 4 weeks, and the remaining 9 followed a VLCKD for 4 weeks.

### 2.2. Dietary Interventions

The VLCKD was characterized by a daily caloric intake of less than 800 kcal, with carbohydrates restricted to less than 50 g per day, protein intake adjusted to 1.2–1.5 g per kilogram of ideal body weight, and fat intake limited to 15–30 g per day. The LCD provided approximately 1000 kcal per day, structured with 65% of total caloric intake from carbohydrates and 35% from fats, excluding protein sources, which was independently calculated and adjusted to 0.8–1 g per kilogram of ideal body weight. All participants were advised to drink 2 L of water/day and to select vegetarian and healthy sources of fat, with protein mainly coming from fish, eggs, fresh dairy products, and lean meat. For the lean controls (N group), a balanced, isocaloric diet was prescribed to maintain body weight, with a macronutrient distribution of 50–55% carbohydrates, 20–25% fats, and 20–25% protein. The potential for micronutrient intake deficiencies was mitigated by providing supplements containing vitamins, minerals, and omega-3 fatty acids, which aligned with international recommendations. Diet adherence was monitored through consultations with trained dietitians every week. Compliance was evaluated at each visit using a 3-day dietary recall, which provided detailed insight into participants’ adherence to the prescribed dietary protocols.

### 2.3. Metabolic, Anthropometric, and Inflammatory Parameters

In patients after an overnight fasting period, blood samples were collected to measure fasting glucose, insulin, CRP, and lipid profile, including triglycerides and total high-density lipoprotein cholesterol (HDL-c). Low-density lipoprotein cholesterol (LDL-c) was calculated using the Friedewald formula. These measurements were used to assess metabolic status and calculate insulin sensitivity using the Homeostatic Model Assessment of Insulin Resistance (HOMA-IR), based on the following formula: Fasting Glucose (mg/dL) × Fasting Insulin (μU/mL)/405.

Routine biochemical tests were conducted according to standard operating practices. Body weight was recorded to the nearest 0.1 kg using a calibrated balance-beam scale (Seca GmbH & Co, Hamburg, Germany). Height was measured to the nearest 0.5 cm using a wall-mounted stadiometer, and body mass index (BMI) was calculated as weight in kilograms divided by height in meters squared. All measurements were performed by trained healthcare professionals to minimize variability and ensure consistency across participants.

### 2.4. Peripheral Blood Mononuclear Cell (PBMC) Isolation and Flow Cytometry Analysis

Peripheral blood samples (4 mL) collected in EDTA tubes were obtained from patients and processed immediately for flow cytometry analysis. Monocytes were identified and analyzed directly in whole blood by staining with specific monoclonal antibodies—anti-HLA-DR-conjugated monoclonal antibody (Miltenyi Biotec, Bergisch Gladbach, Germany), anti-CD14 monoclonal antibody conjugated to Allophycocyanin (APC, 660-20 Red, Miltenyi Biotec), and anti-CD16 monoclonal antibody conjugated to Phycoerythrin (PE, 585-42 Blue, Miltenyi Biotec). After staining, samples were immediately analyzed using an FACS Canto II flow cytometer (FACS Diva Version 6.1.3). Data were acquired from a total of 50,000 events per sample and were further analyzed using FlowJo software v9.3.2. Results are presented as the percentage of each monocyte subpopulation.

### 2.5. Statistical Analysis and Sample Size Calculation

Quantitative variables, including clinical and biological parameters as well as monocyte subset percentages, were expressed as number, mean, and standard deviation, while categorical variables were presented as frequencies and percentages. Statistical analysis was conducted using GraphPad Prism v5.01 (GraphPad Software, San Diego, CA, USA). Group comparisons were performed using unpaired Student’s *t*-test or one-way analysis of variance (ANOVA) for multiple groups, followed by post hoc Tukey’s tests to identify specific group differences. Correlations between monocyte subsets and clinical or metabolic parameters were analyzed using linear regression. The effect of dietary interventions compared to lean and untreated individuals with obesity was evaluated using one-way ANOVA. Statistical significance was set at *p* < 0.05. Based on expected differences in monocyte subset proportions, which were defined as the primary outcome of interest, using preliminary data from previous studies, we estimated the mean and standard deviation of CM percentages in similar populations. Assuming a significance level (α) of 0.05 and a power (1−β) of 0.80, our calculations suggested that detecting a large effect size (35–40% difference in monocyte proportions) would require approximately 10 participants per group.

## 3. Results

### 3.1. Biochemical, Anthropometric, and Metabolic Characteristics of Study Cohorts

A total of 36 participants, including typical-weight individuals and patients with obesity, were enrolled in the study. Table 1 summarizes the anthropometric and biochemical characteristics of the lean subjects (N group; BMI range 19.05–24.8 kg/m^2^) and three groups of individuals with obesity—those without any dietary restrictions (Ob group; BMI range 33.02–44.81 kg/m^2^), those after 4 weeks of a low-calorie diet (LCD group; BMI range 36.8–46.8 kg/m^2^), and those after 4 weeks of a very-low-calorie ketogenic diet (VLCKD group; BMI range 30.79–49.6 kg/m^2^).

Individuals with obesity exhibited significantly impaired glucose metabolism, characterized by elevated fasting glucose, insulin, and insulin resistance, as assessed by HOMA-IR. Notably, the insulinemia and HOMA-IR levels were reduced in the groups following dietary interventions, with the most pronounced improvements observed in the VLCKD group. Additionally, C-reactive protein (CRP) levels were significantly elevated in the Ob and LCD groups compared to the lean group, whereas CRP levels in the VLCKD group were comparable to those of lean participants.

### 3.2. Distribution of Circulating Monocyte Subsets in Typical-Weight Individuals and Those with Obesity

In individuals with obesity who were not following dietary restrictions, the proportion of classical monocytes (CMs) was significantly lower, while intermediate monocytes (IMs) and non-classical monocytes (NCMs) were significantly elevated compared to individuals with normal weight (Figure 1). Specifically, in individuals with obesity and typical-weight individuals, the following values were observed: CMs: 71.67% ± 3.69 vs. 85.65% ± 1.23 (*p* = 0.0012); Ims: 9.60% ± 0.77 vs. 5.73% ± 0.56 (*p* = 0.0005); and NCMs: 13.77% ± 1.52 vs. 5.72% ± 0.82 (*p* = 0.0001).

When analyzing the combined group of typical-weight individuals and those with obesity, with and without dietary interventions, a strong negative correlation was observed between the CM and both the IM and NCM populations. In contrast, the IM and NCM populations positively correlated with each other (Figure 2A–C). Moreover, within this same group, both the IM and NCM populations positively correlated with BMI and CRP levels, while the CM populations demonstrated a negative correlation with these parameters (Figure 3A–C and Figure 4A–C).

### 3.3. Changes in Circulating Monocyte Subsets After Dietary Interventions

After 4 weeks of a VLCKD, a peculiar distribution in monocyte subset distribution was observed. The VLCKD group exhibited a notable reduction in NCMs (8.27% ± 1.17; *p* = 0.005) and IMs (7.50% ± 0.75; *p* = 0.038), alongside an increase in CMs (78.45% ± 1.39; *p* = 0.052) compared to the group of individuals with obesity who did not follow a diet. These changes indicated a trend toward the normal proportions seen in typical-weight individuals, particularly with regard to NCMs, which reached levels closer, although different, to those of the lean group (8.27% ± 1.17 vs. 5.72% ± 2.45; *p* = 0.047).

In contrast, after 4 weeks of a LCD, the distribution of circulating monocyte subsets (CMs: 72.73 ± 3.17%; IMs: 11.08 ± 1.76%; NCMs: 11.06 ± 1.34%) did not show significant differences (*p* = 0.414, *p* = 0.207, and *p* = 0.089, respectively) compared to individuals with obesity who did not undergo dietary restrictions (Figure 5).

## 4. Discussion

To date, studies investigating the distribution of circulating monocyte subsets in individuals with obesity using flow cytometry have produced garbled results, likely due to variations in technical methods and experimental design. Our data demonstrate that the proportion of monocyte subsets is significantly altered in individuals with obesity, with a marked increase in both NCMs and IMs, as well as a corresponding decrease in CMs. These findings are consistent with other chronic inflammatory conditions.

The roles and relationships between monocyte subsets in both physiological and pathological states are highly complex and require further clarification. Monocyte subsets exhibit distinct functions, including pro-inflammatory responses, tissue repair, and immune surveillance, which can vary depending on the context. Functional and gene expression studies have shown that the monocyte subsets in individuals with obesity respond differently to inflammatory and chemotactic stimuli compared to those in lean individuals [7,22]. Recent strands of research have revealed that CMs and IMs in individuals with obesity express more chemotactic molecules, which suggests a greater propensity to migrate into adipose tissue [22]. These authors propose a model in which pro-inflammatory monocytes in individuals with obesity respond to circulating factors via Toll-like receptors (TLR4 and TLR8). In response, CMs and NCMs secrete increased levels of inflammatory cytokines, while the IMs and CMs, which express more CCR5 and CCR2, may have an enhanced ability to migrate into tissues, where they can differentiate into macrophages. This process sustains the low-grade inflammation characteristic of obesity, while NCMs may remain in circulation for longer to perform patrolling functions [22].

Ex vivo studies further highlight the functional differences among the monocyte subsets. CMs demonstrated the highest migratory capacity towards human omental adipose tissue-conditioned media, while IMs and, to a greater extent, NCMs exhibited a significantly reduced migration [23]. These findings emphasize the complexity of monocyte subset functions and their differential roles in both normal and pathological conditions.

Moreover, the proportions of circulating monocyte subsets may reflect the inflammatory state of visceral adipose tissue independently of BMI. An increased proportion of NCMs and IMs, along with a decreased proportion of CMs, may serve as early biomarkers of metabolic dysfunction, appearing even before the onset of clinical symptoms. Targeted therapeutic interventions could potentially normalize monocyte distribution, reflecting improvements in the underlying tissue alterations that contribute to metabolic dysfunction.

Our findings support this hypothesis. We compared the distribution of monocyte subsets in typical-weight individuals and those with obesity, including those on unrestricted diets and those following restricted diets, with a particular focus on the effects of a VLCKD, which is known to have anti-inflammatory properties. To this end, we conducted a retrospective study of individuals with obesity, ensuring that the groups were matched for age and BMI. Despite the short duration of the intervention (4 weeks), there were no significant differences in BMI reduction between the groups following an LCD and those following a VLCKD.

We observed significant differences in monocyte subset distribution only in the VLCKD group, not in the LCD group, compared to individuals with obesity who were not following a diet. Notably, the changes were evident across all subsets, with a trend toward normalization, particularly for NCMs, which approached levels seen in lean individuals. Other studies have shown similar results, with a reduction in NCMs after moderate weight loss following a low-calorie diet in individuals with obesity [16]. Considering the significant correlations identified in our study between BMI and monocyte subsets and given that the two nutritional interventions resulted in comparable BMI and glucose/lipid profiles, we speculate that the observed shifts in monocyte distribution may be influenced by additional factors, such as the nutritional ketosis induced by the VLCKD. Indeed, our data suggest that VLCKDs may directly influence the inflammatory mechanisms associated with obesity by exerting anti-inflammatory effects on visceral adipose tissue. This is reflected in the partial restoration of monocyte subset proportions, which may be associated with subsequent reductions in visceral adipose tissue mass.

Furthermore, our findings align with other studies [22,24] that demonstrate individuals with obesity undergoing a VLCKD experience greater and more sustained weight loss compared to those following an LCD. This effect appears to be mediated by both metabolic mechanisms and the anti-inflammatory properties of the ketogenic diet. Several studies have shown that moderate ketosis modulates inflammation and immune cell function, with β-hydroxybutyrate playing a key role in promoting an anti-inflammatory response [25,26]. Ketone-based metabolic therapy has even emerged as a treatment for attenuating the hyperinflammatory cytokine storm characteristic of severe SARS-CoV-2 infection [27,28,29].

Despite the valuable insights provided by this study, there are several limitations that must be acknowledged. First, the sample size was relatively small, which limits the generalizability of the findings. Additionally, the retrospective design and lack of paired pre- and post-diet data for individual participants restricted our ability to assess within-subject changes in monocyte subsets over time. A prospective study design with longitudinal data would offer stronger evidence of the impact of these dietary interventions. Another key limitation is the absence of β-hydroxybutyrate (BOH) measurements, which are essential for confirming the degree of ketosis in the VLCKD group. BOH levels would have provided direct evidence of the metabolic state induced by the ketogenic diet, helping to clarify the relationship between ketosis and the observed changes in monocyte subsets. Furthermore, we did not measure waist circumference or perform body composition analysis, which are critical for assessing visceral adiposity. Since visceral fat is a major contributor to the inflammation seen in obesity, understanding its relationship with monocyte subset alterations would have added valuable context to our findings. Additionally, the short duration of the dietary interventions may not have been long enough to fully capture the long-term effects of diet on immune cell function and metabolic health. A longer follow-up period would be necessary to evaluate the sustainability of the observed changes.

## 5. Conclusions

Our findings suggest that even a short-term VLCKD intervention can induce significant shifts in the distribution of circulating monocyte subsets, moving from a disrupted pattern to a profile more similar to that of lean individuals. This early impact on monocyte populations may be driven by immune, inflammatory, and metabolic changes in visceral adipose tissue triggered by ketosis. These alterations could play a key role in the long-term benefits and sustained effects of this dietary intervention. However, further studies with larger cohorts and extended follow-up are needed to fully understand the mechanisms and long-term implications of these changes.

## Figures and Tables

**Figure 1 nutrients-17-00312-f001:**
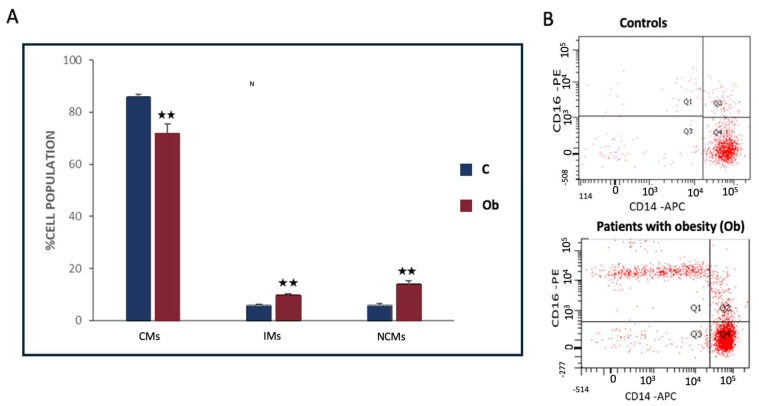
(**A**,**B**) Differences in circulating monocyte subsets between individuals with obesity without dietary restrictions (Ob) and typical-weight controls (N). (**A**) The percentages of classical monocytes (CMs: CD14+CD16−), non-classical monocytes (NCMs: CD14dimCD16+), and intermediate monocytes (IMs: CD14+CD16++) are shown as a proportion of total CD14+ cells (mean ± SD). (**B**) Representative flow cytometry plots illustrating the expression of CD14 and CD16 in both N and Ob groups. Group comparisons were made using one-way ANOVA. ⋆⋆: *p* < 0.001.

**Figure 2 nutrients-17-00312-f002:**
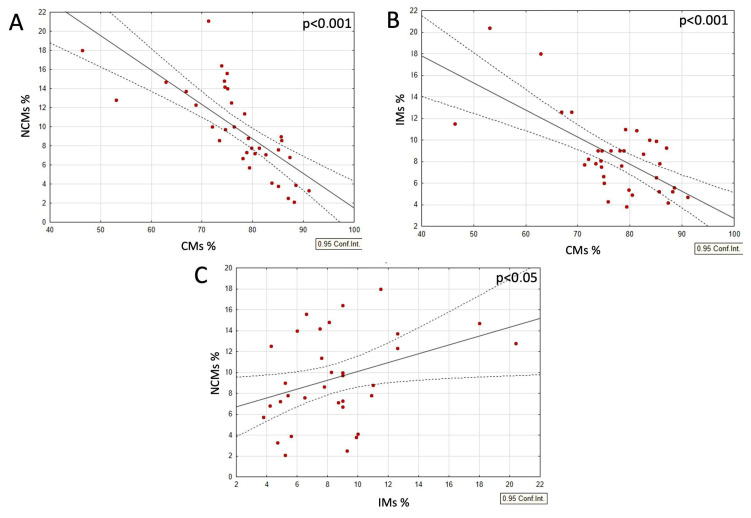
(**A**–**C**) Correlations between subsets of circulating monocytes in all participants. (**A**) Correlation between classical monocytes (CMs) and non-classical monocytes (NCMs), (**B**) CMs and intermediate monocytes (IMs), and (**C**) NCMs and IMs. *p*-values were obtained through simple linear regression analysis, with the following results: (**A**) CMs vs. NCMs, *p* < 0.001; (**B**) CMs vs. IMs, *p* < 0.001; (**C**) NCMs vs. IMs, *p* < 0.05.

**Figure 3 nutrients-17-00312-f003:**
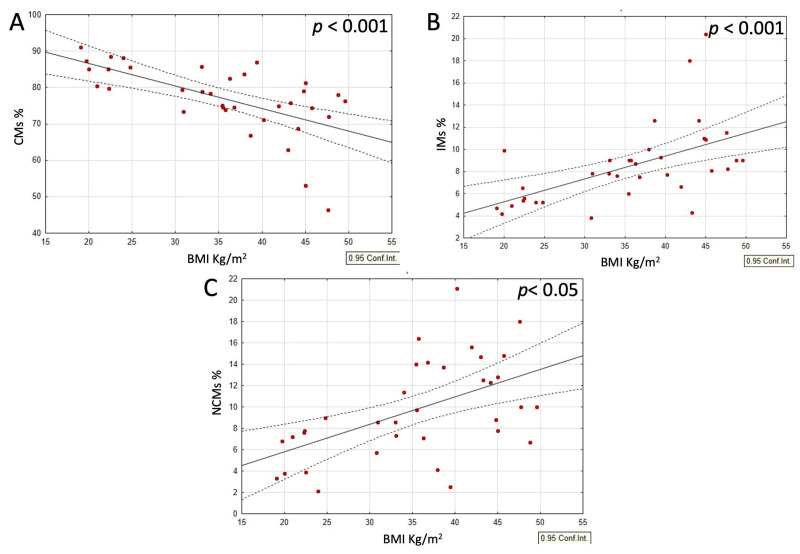
(**A**–**C**) Correlations between subsets of circulating monocytes and body mass index in all participants. (**A**) Correlation between CMs and BMI, (**B**) IMs and BMI, and (**C**) NCMs and BMI. *p*-values were obtained through simple linear regression analysis, with the following results: (**A**) *p* < 0.001; (**B**) *p* < 0.001; (**C**) *p* < 0.05.

**Figure 4 nutrients-17-00312-f004:**
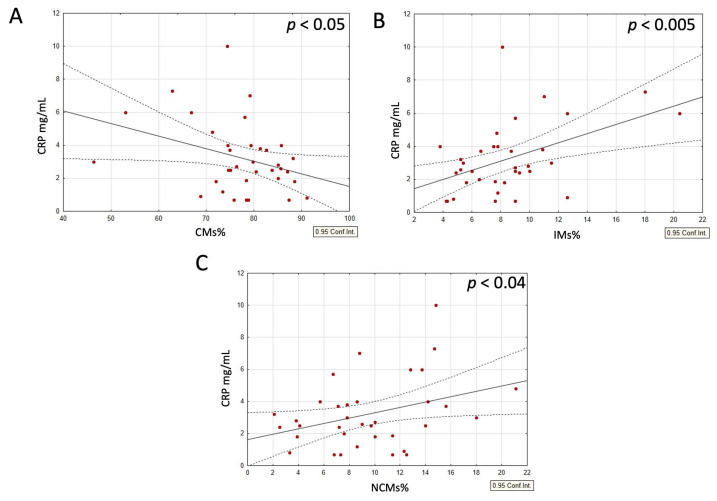
(**A**–**C**) Correlations between CRP and subsets of circulating monocytes in all participants. (**A**) Correlation between CRP and CMs, (**B**) CRP and IMs, and (**C**) CRP and NCMs. *p*-values were obtained through simple linear regression analysis, with the following results: (**A**) *p* < 0.05; (**B**) *p* < 0.005; (**C**) *p* < 0.04.

**Figure 5 nutrients-17-00312-f005:**
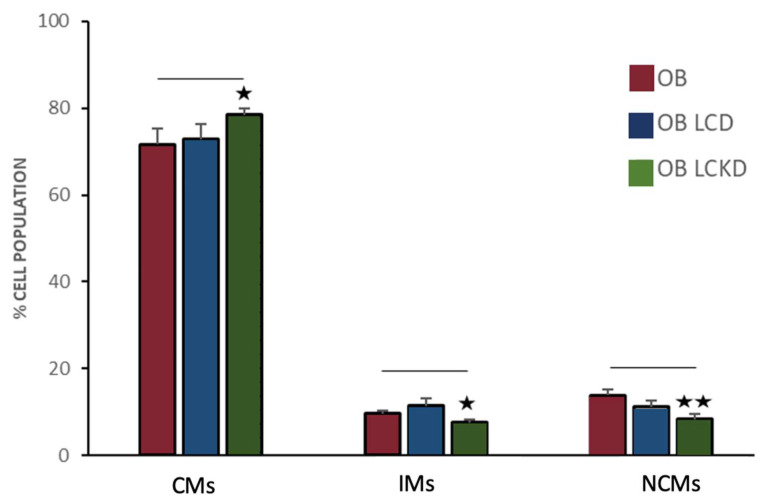
Changes in monocyte subsets in individuals with obesity after 1 month without dietary intervention (OB), following a low-calorie diet (LCD), and after a very-low-calorie ketogenic diet (VLCKD). Group comparisons were performed using one-way ANOVA. Statistically significant differences are indicated as ⋆: *p* = 0.05, ⋆⋆: *p* = 0.005.

**Table 1 nutrients-17-00312-t001:** Clinical and biochemical characteristics of study participants across different groups.

Parameter	Lean Controls (N)	Patients with Obesity (OB)	Patients with Obesity After a Low-Calorie Diet (LCD)	Patients with Obesity After a Very-Low-Calorie Ketogenic Diet (VLCKD)	*p*-Value (N vs. OB, LCD, VLCKD)
Number of subjects	9	9	9	9	
Sex ratio (F/M) (%)	5/4 (55.5%)	5/4 (55.5%)	5/4 (55.5%)	5/4 (55.5%)	
Age (years)	40.56 ± 2.65	41.40 ± 3.13	40.87 ± 3.13	41.18 ± 1.70	0.33-0.41-0.45
Weight (kg)	62.72 ± 3.51	119.49 ± 7.40	112.78 ± 7.31	114.30 ± 10.98	**<0.0001**
BMI (kg/m^2^)	21.73 ± 0.65	40.01 ± 1.69	41.18 ± 1.70	38.62 ± 2.39	**<0.0001**
Fasting glycemia (mmol/L)	4.68 ± 0.18	5.52 ± 0.36	5.29 ± 0.23	5.28 ± 0.25	**0.026-0.024-0.04**
Insulinemia (μU/mL)	9.83 ± 0.53	20.53 ± 4.83	16.79 ± 1.63	15.79 ± 2.98	**0.022-0.0004-0.033**
HOMA-IR	2.03 ± 0.11	5.32 ± 1.27	3.96 ± 0.41	3.64 ± 0.66	**0.010-0.0002-0.014**
Total C (mmol/L)	4.51 ± 0.32	4.75 ± 0.22	4.36 ± 0.25	4.41 ± 0.18	0.294-0.375-0.404
HDL-C (mmol/L)	1.16 ± 0.06	1.20 ± 0.06	1.16 ± 0.09	1.23 ± 0.13	0.320-0.484-0.320
LDL-C (mmol/L)	2.71 ± 0.20	3.06 ± 0.25	2.60 ± 0.18	2.59 ± 0.17	**0.05-0.038-0.04**
Triglycerides (mmol/L)	1.40 ± 0.18	1.07 ± 0.17	1.31 ± 0.18	1.30 ± 0.15	0.118-0.388-0.335
CRP (mg/L)	2.14 ± 0.69	5.37 ± 1.75	5.42 ± 1.25	2.62 ± 0.55	**0.044-0.01**-0.229
Neutrophil count (×10^9^/L)	4.21 ± 0.34	4.52 ± 0.23	4.82 ± 0.45	4.17 ± 0.36	0.121-0.233-0.467
Lymphocyte count (×10^9^/L)	2.24 ± 0.24	2.41 ± 0.20	1.91 ± 0.17	1.97 ± 0.19	0.351-0.130-0.197
Monocyte count (×10^9^/L)	0.48 ± 0.03	0.51 ± 0.20	0.53 ± 0.04	0.47 ± 0.05	0.218-0.167-0.192

**Abbreviations:** lean controls (N); individuals with obesity (OB); individuals with obesity after a low-calorie diet (LCD); and individuals with obesity after a very-low-calorie ketogenic diet (VLCKD). BMI: body mass index; F: female; M: male; HOMA-IR: homeostatic model assessment for insulin resistance; Total C: total cholesterol; HDL-C: high-density lipoprotein cholesterol; LDL-C: low-density lipoprotein cholesterol; CRP: C-reactive protein. *p*-values were obtained using one-way ANOVA for comparisons between groups (N vs. OB, LCD, VLCKD). Statistically significant *p*-values (*p* < 0.05) are shown in bold.

## Data Availability

Data will be made available upon reasonable request to the corresponding author.

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
