# Peer review of "The Impact of a Very-Low-Calorie Ketogenic Diet on Monocyte Subsets of Patients with Obesity: A Pilot Study"

_nutrients, 2025, doi:10.3390/nu17020312_

Round 1

Reviewer 1 Report

Comments and Suggestions for Authors

In general, the manuscript was prepared correctly. I would like to draw attention to the lack of statistical methods to assess the differences between the 3 groups (Table 1) in relation to the control. There is also a lack of research on the effect of the type of diet and obesity. Did the authors estimate the sample size? How was the intake monitored? Were supplements given in the low-calorie diets? The description only includes findings for macronutrients, and how were the others balanced with such a caloric content. Due to the lack of data before the intervention and the lack of control of the ketosis state (which was included as a limitation), I propose that in the title and text define the study as pilot and change the category from article to communication. The authors' observations seem to be preliminary and there is a lack of basic data to consider the results and conclusions as certain.

Author Response

Comment 1: In general, the manuscript was prepared correctly. I would like to draw attention to the lack of statistical methods to assess the differences between the 3 groups (Table 1) in relation to the control. 
Response 1:
Thank you for your comment. To analyze the differences between the three groups and the control group in Table 1, one-way ANOVA was employed for multiple group comparisons, followed by post-hoc Tukey's tests to identify specific group differences. These details have been added to the "Statistical Analysis" section, and p-values have been included in Table 1 for clarity.

Comment 2: Lack of research on the effect of the type of diet and obesity.
Response 2:
We have expanded the discussion to include additional context on the impact of diet type (low-calorie versus ketogenic) and obesity on inflammation and immune function. Relevant literature has been cited to provide a more comprehensive understanding of the observed effects. These revisions enhance the integration of our findings with existing knowledge.

Comment 3: Did the authors estimate the sample size?
Response 3:
Thank you for your comment. We conducted an initial sample size calculation based on expected differences in monocyte subset proportions, which were defined as the primary outcome of interest. Using pilot data from previous literature, we estimated the mean and standard deviation of classical monocyte (CM) percentages in similar populations. Assuming a significance level (α) of 0.05 and a power (1−β) of 0.80, our calculations suggested that detecting a large effect size (35–40% difference in monocyte proportions) would require approximately 10 participants per group.

Comment 4: How was the intake monitored? Were supplements given in the low-calorie diets?
Response 4:
We have clarified in the revised manuscript that dietary adherence was monitored through weekly dietary recalls by expert dieticians (SB and EC). Additionally, participants in both dietary intervention groups received multivitamin and mineral supplements to prevent deficiencies, particularly in the low-calorie ketogenic diet group. These details have been added to the "Dietary Interventions" section.

Comment 5: The description only includes findings for macronutrients, and how were the others balanced with such a caloric content?
Response 5:
We have updated the "Dietary Interventions" section to provide more information regarding the low-calorie and the ketogenic nutritional interventions. Supplementation with vitamins and minerals was provided to ensure adequate intake, and participants were instructed to consume a variety of nutrient-dense foods within the prescribed macronutrient limits.

Comment 6: Due to the lack of data before the intervention and the lack of control of the ketosis state (which was included as a limitation), I propose that in the title and text define the study as pilot and change the category from article to communication.
Response 6:
We agree with the reviewer’s suggestion. The title has been revised to reflect the pilot nature of the study, now reading: " Impact of a Very Low-Calorie Ketogenic Diet on Monocyte Subsets in Obesity: a pilot study." Additionally, the manuscript has been revised to describe the study as preliminary in nature, and the submission category has been updated to "Communication," pending editorial approval.

Comment 7: The authors' observations seem to be preliminary, and there is a lack of basic data to consider the results and conclusions as certain.
Response 7:
We acknowledge the preliminary nature of the study and have made revisions to the discussion and conclusion sections to emphasize this point. Specifically, we have highlighted the limitations related to the small sample size and  retrospective design, while framing the results as a foundation for future, more robust studies. This adjustment aligns with the reviewer's perspective and ensures a balanced interpretation of the findings.

Reviewer 2 Report

Comments and Suggestions for Authors

This study investigated the effect of a very low-calorie ketogenic diet (VLCKD) on patients with obesity. The authors found that CRP levels were restored following VLCKD treatment in obese patients. Moreover, the reduction of classical monocytes and the increase in non-classical monocytes and intermediate monocytes observed in obesity were normalized by VLCKD treatment. The authors’ claims are supported by the data, and the findings are meaningful because the effects of VLCKD treatments were observed in clinical trials. However, the following points need to be addressed:

1.       In Figures 2, 3, and 4, the letters are too small to read.

2.       In Table 1, the row labeled "Lean controls (C)" should be corrected to "Lean controls (N)."

3.       In the Discussion, the authors state, “CM demonstrated the highest migratory capacity towards human omental adipose tissue-conditioned media, while IM and, to a greater extent, NCM exhibited significantly reduced migration (Pecht, 2016).” Is (Pecht, 2016) a reference? If so, it should be included in the References section.

Author Response

Comment 1: 

This study investigated the effect of a very low-calorie ketogenic diet (VLCKD) on patients with obesity. The authors found that CRP levels were restored following VLCKD treatment in obese patients. Moreover, the reduction of classical monocytes and the increase in non-classical monocytes and intermediate monocytes observed in obesity were normalized by VLCKD treatment. The authors’ claims are supported by the data, and the findings are meaningful because the effects of VLCKD treatments were observed in clinical trials. However, the following points need to be addressed:

In Figures 2, 3, and 4, the letters are too small to read.

Response 1:
Thank you for your kind comment. We have revised Figures 2, 3, and 4 to ensure that all text, including axis labels, legends, and annotations, is clearly legible. The updated figures have been enlarged, and the font size has been increased for clarity. These revised figures are included in the updated manuscript.

Comment 2:
In Table 1, the row labeled "Lean controls (C)" should be corrected to "Lean controls (N)."
Response 2:
We appreciate your attention to this inconsistency. The label in Table 1 has been corrected from "Lean controls (C)" to "Lean controls (N)" to align with the rest of the manuscript. This correction has been implemented in the revised manuscript.

Comment 3:
In the Discussion, the authors state, “CM demonstrated the highest migratory capacity towards human omental adipose tissue-conditioned media, while IM and, to a greater extent, NCM exhibited significantly reduced migration (Pecht, 2016).” Is (Pecht, 2016) a reference? If so, it should be included in the References section.
Response 3:
Thank you for catching this oversight. (Pecht, 2016) is indeed a reference. We have verified its citation and added the full reference to the References section as follows:
Pecht, T.; Gutman-Tirosh, A.; Bashan, N.; Rudich, A. Peripheral Blood Leucocyte Subclasses as Potential Biomarkers of Adipose Tissue Inflammation and Obesity Subphenotypes in Humans. Obesity Reviews, 2014, 15, 322–337, doi:10.1111/obr.12133.
This correction has been made in the revised manuscript.

Reviewer 3 Report

Comments and Suggestions for Authors

The study by Curreli et al. aimed to assess the effect of a very low-calorie ketogenic diet on monocyte subset distribution among patients with obesity. The study's topic is interesting and undoubtedly of interest to clinicians.

However, I have doubts about whether the study should be published. Many issues with the methodology and results question the study's credibility.
Below are my comments.

MAJOR:

2.1. Study design and population
Point 1. There is no information about sample size estimation
Point 2. Sample size – is the number of subjects included in the study enough to reach adequate power?

2.2. Dietary interventions:
Point 3. Authors wrote: "The LCD provided 1000 kcal per day, structured with 65% of total caloric intake from carbohydrates and 35% from fats (excluding protein sources), with protein intake adjusted to 0.8–1 g per kilogram of ideal body weight."
When 65% are CHO, 35% are fats, so there was no protein in the diet? (65+35 = 100%).
Point 4. There is no information about diet specifications in the lean group (what does a balanced diet mean?)

2.3. Metabolic, Anthropometric and Inflammatory Parameters
Point 5. the descriptions in this section are inaccurate and insufficient, e.g.:
LDL-c was calculated od measured by direct methods?
Height was measured to the closed 0,5 cm, how about weight?
No references eg. to HOMA-IR formula
no information about BMI in methods (but in table 1 BMI is presented)

Point 6.
Lack of assessment of dietary compliance or lack of comparison of the nutritional value of the diet between the studied groups is a serious problem.

Point 7.
In the results, there are abbreviations concerned groups eg. C-group (but in table 1 - N-group?) or Ob.-grup. The abbreviations should be introduced and explained before (in 2.1 section)
The same - In Table 1 – Lean controls (c) but in abbreviations below Table 1 – Lean controls (N)
Point 8.
Table 1 – why LDL concentration is not included in Table 1?

Point 9.
If the shift in monocyte distribution was affected by body mass reduction (% weight loss) but not by the model of diet. There is no information about changes in body mass, BMI, and glucose and lipid parameters during the intervention. Perhaps the change in these parameters is, to a greater extent, related to the observed changes in the distribution of monocytes than the reduction diet model (LCD, VLCKD).
Is there Any additional information about it in the supplementary material?

Point 10
page 8, second paragraph, line 4 -> incorrect citation style
Page 8, third paragraph -> There is no citation; please provide adequate citations to support the presented information.

Point 11
Study limitations
The authors correctly identified the study's limitations. The small size of the study group, the lack of sample size estimation, the lack of supervision over diet compliance, and data on changes in body composition during the study are huge barriers to the recognition of the work as valuable.

Minor suggestions::
page 1, line 6 -> "[1], [2]." change into [1, 2] (the same eg. on page 2, line 11, and through entire manuscript)

Author Response

Comment 1: There is no information about sample size estimation.
Response 1:
Thank you for your comment. We conducted an initial sample size calculation based on expected differences in monocyte subset proportions, which were defined as the primary outcome of interest. Using pilot data from previous literature, we estimated the mean and standard deviation of classical monocyte (CM) percentages in similar populations. Assuming a significance level (α) of 0.05 and a power (1−β) of 0.80, our calculations suggested that detecting a large effect size (35–40% difference in monocyte proportions) would require approximately 10 participants per group.

Comment 2: Is the number of subjects included in the study enough to reach adequate power?
Response 2:
Given the exploratory nature of this study, the number of patients included is in line with a preliminary sample size estimation. However, we recognize that the small sample size may limit the statistical power and the ability to generalize our findings. This has been clarified in the discussion. We have also emphasized that the results should be interpreted as preliminary and as a foundation for larger, more robust studies.

2.2 Dietary Interventions

Comment 3: "65% are CHO, 35% are fats, so there was no protein in the diet?"
Response:
We apologize for the confusion. The percentages of macronutrients referred to carbohydrates and fats excluding protein sources. We have further clarified this in the revised text, explicitly stating that protein intake was calculated separately and adjusted to 0.8–1 g per kilogram of ideal body weight.

Comment 4: There is no information about diet specifications in the lean group (what does a balanced diet mean?)
Response 4:
We have added a detailed description of the diet followed by the lean group, specifying it included an isocaloric diet designed to maintain body weight, with a macronutrient distribution of 50–55% carbohydrates, 20–25% fats, and 20–25% protein. This information has been incorporated into the “Dietary Interventions” section.

2.3 Metabolic, Anthropometric, and Inflammatory Parameters

Comment 5: Descriptions in this section are inaccurate and insufficient.
Response 5:
The following clarifications and additions have been made in the revised manuscript:

  • LDL-c: We specified that LDL-c was calculated using the Friedewald formula.
  • Weight: We confirmed that weight was measured using a calibrated balance-beam scale with readings to the nearest 0.1 kg.
  • HOMA-IR formula: The reference for the HOMA-IR formula has been included.
  • BMI: Information on how BMI was calculated (weight in kilograms divided by height in meters squared) has been added to the "Methods" section.

Comment 6: Lack of assessment of dietary compliance or comparison of the nutritional value of the diet between groups.
Response 6:
Dietary compliance was partially assessed through weekly food diaries and recalls, which has now been mentioned in the “Dietary Interventions” section. However, a detailed comparison of the nutritional value of diets between groups was not performed, which we will be addressed in future studies. We have acknowledged this limitation and have highlighted it in the “Study Limitations” section.

Comment 7: Abbreviation inconsistencies (e.g., C-group, N-group, Ob-group).
Response 7:
We appreciate the reviewer pointing out these inconsistencies. All abbreviations have been standardized throughout the manuscript. Specifically, “N-group” (Lean controls) is used consistently in the text, figures, and tables.

Comment 8: Why LDL concentration is not included in Table 1?
Response 8:
Thank you for this suggestion. LDL concentration has now been added to Table 1 in the revised manuscript.

Comment 9: The shift in monocyte distribution may be related to changes in body mass, BMI, and glucose/lipid parameters rather than diet model. Is there any additional information in supplementary material?
Response 9:
Thank you for your observation. Considering the significant correlations identified in our study between BMI and monocyte subsets, and given that the two nutritional interventions resulted in comparable BMI and glucose/lipid profiles, we speculate that the observed shifts in monocyte distribution may be influenced by additional factors, such as the nutritional ketosis induced by the Very Low-Calorie Ketogenic Diet. Ketosis is known to exert anti-inflammatory effects and modulate immune cell functions, which may explain the differences observed in monocyte subsets. These findings underscore the importance of future research to specifically explore the role of ketosis in immune modulation, as discussed in our paper.

Comment 10: Incorrect citation style and lack of citations in some paragraphs on page 8.
Response 10:
We have corrected the citation style throughout the manuscript to use consistent formatting (e.g., [1, 2] instead of "[1], [2]."). Missing citations have been added in the relevant paragraphs on page 8 to support the statements.

Comment 11: Study limitations are significant.
Response 11:
We agree with the reviewer and have expanded the “Study Limitations” section to acknowledge the small sample size, limited supervision of dietary compliance, and lack of body composition analysis. We have emphasized that these limitations restrict the generalizability and interpretability of the findings.

MINOR SUGGESTIONS

Page 1, line 6: "[1], [2]." change to [1, 2] (and throughout the manuscript).
Response:
This correction has been implemented throughout the manuscript for consistency.

Round 2

Reviewer 1 Report

Comments and Suggestions for Authors

I greatly appreciate the authors' meticulous corrections to their manuscripts. There are no comments on this revised version

Reviewer 3 Report

Comments and Suggestions for Authors

Thank you for your responses to my suggestions. Extensive changes have been made to the manuscript. Some issues regarding the methodology, such as a small number of subjects, lack of a broad quantitative assessment of the diet are still present but I understand the inability to change them, which has been included in the manuscript. I believe that in future research the authors will take into account the experiences gained from previous work. I accept the manuscript for submission.